# How does climate change affect potential yields of four staple grain crops worldwide by 2030?

**Chengzhi Cai**[1]*, **Linyu LV**[1,2], **Sha WEI**[1,2], **Lin ZHANG**[1], **Wenfang CAO**[3]

**1** Economic Institute, Guizhou University of Finance and Economics, Guiyang, China, **2** School of Foreign Languages, Guizhou University of Finance and Economics, Guiyang, China, **3** School of Environmental Sciences and Engineering, Southern University of Science and Technology, Shenzhen, China

* caichengzhi@mail.gufe.edu.cn

**Data Availability Statement:** All relevant data are within the paper and its Supporting Information files.

**Funding:** The author(s) received no specific funding for this work.

## Abstract

Global food security basically depends on potential yields of staple grain crops worldwide, especially under climate change. However, most scholars use various models of production function in which climatic factors are often considered to estimate crop yield mostly at local or regional level. Therefore, in this paper: Potential yields of rice, wheat, maize and soybean worldwide by 2030 are projected creatively using Auto-regressive Integrated Moving Average and Trend Regressed (ARIMA-TR) model in which actual yields in recent two years are used for testing the reliability of projection and Gray System (GS) model for validating the test; Especially individual impacts of climate change on the productions of rice, wheat, maize and soybean worldwide since 1961 are analyzed by using unary regression model in which global mean temperature and land precipitation are independent variable while the yield of crop being dependent one, respectively. Results show that: by 2030, the ratio between average and top yields of world rice is projected to be 50.6% increasing, while those of world wheat, world maize and world soybean are projected to be 38.0% increasing, 14.7% decreasing and 72.5% increasing, respectively. Since 1961 global warming has exerted a negative impact on average yield of world rice less than on its top, a positive effect on average yield of world wheat while a negative impact on its top, a positive effect on average yield of world maize less than on its top, and a positive influence on average yield of world soybean while a negative one on its top, which might be slightly mitigated by 'Carbon Peak' target. The fluctuation of global rainfall contributes to the productions of these crops much less than global warming during same period. Our findings indicate that: to improve global production of four staple grain crops by 2030, the priorities of input should be given to either rice or wheat in both high and low yield countries, whereas to maize in high yield countries and to soybean in low yield countries. These insights highlight some difference from previous studies, and provide academia with innovative comprehension and policy-decision makers with supportive information on sustainable production of these four staple grain crops for global food security under climate change in the future.

**Competing interests:** The authors have declared that no competing interests exist.

## Introduction

The UN's Sustainable Development Goals (SDG) prioritize global food security as a critical agenda for the year 2030. The attainment of such a goal hinges significantly upon the production of staple grain crops. Concurrently, the ongoing challenge of global warming, which is projected to persist until at least 2030 under the 'Carbon Peak' target set forth in *The Paris Agreement* of 2016, further underscores the importance of studying potential yields of staple grain crops, particularly in the context of climate change. Recent scholarly investigations into model-estimated potential yields of such staple grain crops as rice, wheat, maize and soybean, especially under climate change, have yielded noteworthy findings as follows.

Rice and wheat, as two most important cereal crops in the world, attract huge scholarly attentions for improving their potential yields. The accuracy and efficiency of optimization algorithms was assessed by comparing the POWELL and SCE-UA method to predict regional winter wheat yield [1], and the target yield for super hybrid rice production was determined by average yield method or regression model method [2]. The correlation between twenty one years (1990–2010) weather data and reported rice yield was used for simulating the yield in the districts of eastern and northeastern region of India under various management conditions [3], while the CERES-Wheat model was employed to estimate regional production of wheat in India's Bihar [4]. The ORYZA rice crop model was used for estimating yield potential of irrigated rice for two widely adopted rice varieties (M-206 and CXL745) in three major US rice-producing regions [5], whereas a linear mixed-effect model was presented to predict wheat yield in northern grain-growing region of Australia [6]. Regional area-weighted mean yield potential (YP) levels of rice in northeastern China (NEC) were estimated by ORYZA (v3) crop model [7], and an improved Carnegie-Ames-Stanford approach (CASA) model was coupled with time-series satellite remote sensing images to estimate winter wheat yield in the country [8]. A rice yield estimation system was developed by using the crop growth model ORYZA and SAR-derived key information to show spatial distribution of rice yield estimates in southern and southeastern Asian countries [9], and a light use efficiency model (EC-LUE) was used for estimating winter wheat yield in Kansas of USA [10]. Wang et al., (2020) presented an improved remote sensing approach to assess regional rice yield gap (YG) in northeastern China during the period from 2006 to 2017 [11], and Liu et al. (2021) explored data assimilation-crop modeling strategies to estimate the yield of winter wheat under different water stress conditions across different growing areas in the country [12]. Sun et al. (2021) evaluated 14 rice growth models against four year phytotron experiments with four levels of heat treatments [13], while Qiao et al. (2021) used optimality-based models for simulating the impacts of climate impacts on global potential wheat yield [14]. Islam et al. (2022) applied multiple statistical models for assessing the variability of climate-induced rice yields in northwest Bangladesh [15], while Gunawat et al. (2022) used the DSSAT model for assessing the impact of climate change on wheat yield in the semi-arid environment [16], and Asis et al. (2023) investigated the impact of climate change on future productivity and water use efficiency of wheat in eastern India [17].

As the most important animal fodder crop with the largest global production among cereals in the world, maize certainly has been attracting scholarly attention for improving its potential yield. A well parameterized and validated Agricultural Production Systems IMulator (APSIM) model was used for assessing the productivity and yield gap of maize in Madhya Pradesh of India [18], the Food and Agriculture Organization (FAO) AquaCrop model was evaluated for its predictability potential of maize growth and yields in Uganda [19], and the AquaCrop and AgroC models were calibrated and validated by using the data sets from 2015 (cool/dry season) and 2016 (warm/wet) respectively to investigate maize development and suitability in cool

climate in Lithuanian of USA [20]. The AquaCrop model was evaluated relatively to maize growth, yield, and water use parameters/variables under different water stress conditions over six years (2005–2010) in Nebraska of USA [21], while the Environmental Policy Integrated Climate (EPIC) model was assessed for its potential to simulate maize yield using limited data from field trials on maize in the eastern Cape of South Africa [22]. Three crop simulation models (AEZ-FAO, DSSAT-CERES-Maize and APSIM-Maize) were calibrated and evaluated by using weather, soil and maize yield data from 79 experimental sites in Brazil to estimate maize potential and attainable yields and to assess the performance of different ensemble strategies to reduce their uncertainties for maize yield prediction [23], and a simple approach to assimilating Moderate Resolution Imaging Spectroradiometer (MODIS) products into a crop growth model was developed to evaluate regional yield prediction performance in Illinois state of USA [24]. Zhang et al. (2019) used different climatic crop models for evaluating Chinese yield loss of maize in the future [25], while Yin and Leng (2020) found that climate variability controls 42.0% of global maize yield variations from 1980 to 2010 and historical climate trend leading to the yield loss by 1.5–3.8% during the period 1980–1990 in China [26]. Zhu et al. (2021) first used hierarchical linear modeling (HLM) method for predicting maize yield in Jilin Province of China [27], Xu et al. (2022) put forward that the detrimental effect of global warming on maize yield might reduce the capacity of bio-energy with carbon capture and storage (BECCS) and threaten food security in the future [28], and Kothiyal et al. (2023) modeled the climate change impact of mitigation (RCP 2.6) and high emission (RCP 8.5) scenarios on maize yield and possible adaptation measures in different agroclimatic zones of Punjab, India [29].

Soybean that produces seed mainly containing protein and oil, as a legume crop of nutritionally rich food as well as feed in the world, attracts the attention of academia to increase its potential future yields. The CROPGRO-Soybean model was used for projecting yield of soybean in Madhya Pradesh of India [30], whereas the Deep CNN-LSTM Model for predicting soybean yield at county-level of China [31]. Using the AquaCrop model, Tovjanin et al. (2019) estimated the impact of climate change on main field crops (maize and soybean) in the Republic of Serbia for the 2041–2070 and 2071–2100 periods [32], whereas Fuzzo et al. (2020) proposed a new method for simulating soybean yield in Parana state of Brazil for 2002–2003 to 2011–2012 [33]. A mathematical model was used for estimating the dependence of soybean seed yield losses upon cutting height variation [34], while soybean Simulation Model (SSM-iCrop2) and GIS were adopted to determine potential yield and the yield gap of soybean in Golestan Province of Iran [35]. The CROPGRO-Soybean was used for assessing soybean yield variability in the southeastern US [36] and for quantifying potential yield and yield gaps of soybean in the humid tropics of southwestern Ethiopia [37], and an improved deep learning model was applied to recognize the prediction of soybean yield [38]; etc.

As discussed above, while numerous research reports exist on model-estimated potential yields of such staple grain crops as rice, wheat, maize and soybean under climate change, most of them are grounded in the principle of production function, focusing on specific variety of crop, adopting static biological perspective, and providing advisory service at local or regional level. Namely, a significant research gap exists regarding the adoption of dynamic evolutionary perspective and global-scale analysis using time-series model for generic crops worldwide. To address such a gap, this paper presents a novel approach leveraging historical performance of crop production since 1961 and applying Auto-regressive Integrated Moving Average and Trend Regressed (ARIMA-TR) model, to project average and top (national) yields of rice, wheat, maize and soybean worldwide by 2030, respectively. The accuracy of these projections is rigorously evaluated through their comparisons with actual yields in recent two years and further validated by using Gray System (GS) model. Especially, unary regression models are employed to assess individual impacts of global warming with 'Carbon Peak' concept and land

precipitation on average and top (national) yields of four staple grain crops since 1961, respectively. This research aims to offer innovative insights to academic community and provide policymakers with supportive information on directing sustainable production of these four staple grain crops for global food security facing the challenge of climate change.

### Main contributions

The main contributions of this work are as follows.

- Potential yields of four staple grain crops worldwide by 2030 are projected creatively using time-series models;

- Different potential limits of average yields of four staple grain crops worldwide are innovatively suggested;

- Individual impacts of climate change on different yields of four staple grain crops worldwide are revealed.

## Materials and methods

### Materials

Top national yield of specific crop per unit acreage can be considered as potential limit of its average worldwide. To explore individual gap between attainable yield and potential limit of four staple grain crops in the future, average and top (national) yields of rice, wheat, maize and soybean worldwide from 1961 to 2019 are used to project their futures by 2030 while actual yields in 2020 and 2021 are used for testing the reliability of projection, respectively. Particularly, global mean temperatures and land precipitations since 1961 are employed to analyze individual impacts of climate change on these four staple grain crops to yield during same period.

In Fig 1, 'average yield' refers to average yield of certain crop worldwide while 'top yield' comes from a specific country with the yield of same crop topping among its kinds in the world in given years in the following. Top yields of world rice are in such details: Australia in 1962, 1965 to 1969, 1971, 1973, 1989, 1991, 1998 to 1999, 2001, 2010, 2013 to 2019, and 2020; Dominica in 1992; Egypt in 1996 to 1997, 2000, 2002, 2004 to 2009, 2011 to 2012, and 2021; North Korea in 1993; Puerto Rico in 1961, 1963 to 1964, 1970, 1972, and 1974 to 1983; Swaziland in 1984 to 1988, and 1990; Syria in 1994 to 1995; and Uzbekistan in 2003. Top yields of world wheat are in such details: Belgium in 1997 and 2009; Denmark in 1961, 1965 to 1966, 1968, and 1975; France in 1972; Ireland in 1969, 1987 to 1988, 1990 to 1991, 1996, 1999 to 2002, 2004, 2006, 2011, 2014 to 2017, 2019, and 2021; Luxembourg in 1998; Netherlands in 1962 to 1964, 1967, 1970 to 1971, 1973 to 1974, 1976 to 1986, 1989, 1992 to 1995, 2003, 2005, and 2010; New Zealand in 2007, 2012 to 2013, 2018, and 2020; and Zambia in 2008. Top yields of world maize are thus detailed: France in 1961 to 1968; Greece in 1983 to 1984; Israel in 1985 to 1987, 1990, and 2010 to 2011; Jordan in 1992, 2002 to 2003, 2005, 2019, and 2020; Kuwait in 1981, 1988 and 1996; Netherlands in 1977 to 1978; New Zealand in 1969 to 1976, 1979 to 1980, and 1982; Qatar in 1989; Saint Vincent and the Grenadines in 2012 and 2021; and United Arab Emirates in 1991, 1993 to 1995, 1997 to 2001, 2004, 2006 to 2009, and 2013 to 2018. And top yields of world soybean are so detailed: Canada in 1961; Egypt in 2003; Ethiopia PDR in 1970 to1972, 1979 to 1983, 1985, and 1987 to 1989; Italy in 1967, 1973, 1974, 1977, 1984, 1986, 1989 to 1999, 2001, 2002 and 2005; Mexico in 1962, 1963 and 1965; New Zealand in 1975, 1976 and 1978; Paraguay in 1964, 1966, 1968 and 1969; Switzerland in 2000; and Turkey in 2004, and 2006 to 2021.

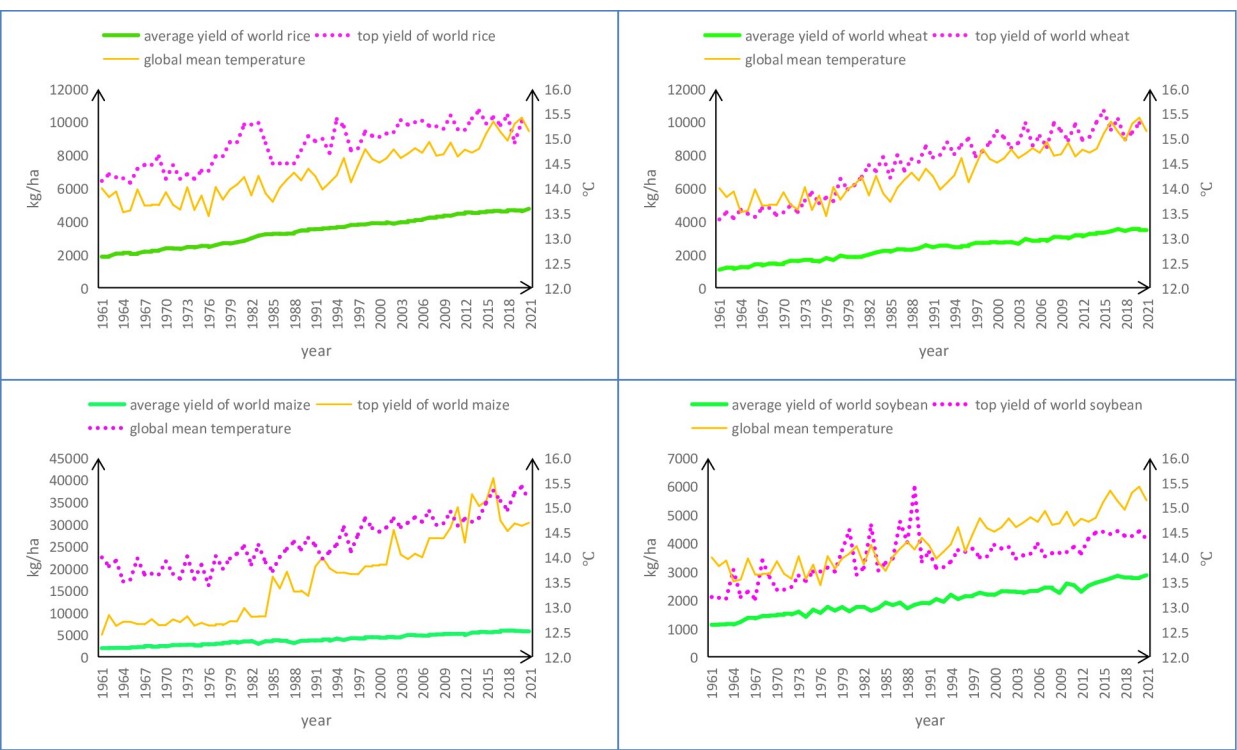

**Fig 1. Average and top yields (kg/ha) of rice, wheat, maize and soybean worldwide, and global mean temperature (˚C) from 1961 to 2021.**
Source: http://www.fao.org/faostat/en/#data/QC; https://www.ncdc.noaa.gov/temp-and-precip/.

Annual land precipitations of the earth since 1961 are not shown in Fig 1 as their variation does not show obvious tendency of increase or decrease. All data used are shown in S1 Table of S1 File.

## Methods

**ARIMA-TR model.** ARIMA-TR model integrates Auto-regressive Integrated Moving Average (ARIMA) model with Trend-regressed (TR) model.

ARIMA model is belonging to time-series approach based on the theory of stochastic process, whose complete representation is mathematically written as Formula (1).

$$[1 - \sum_{i=1}^{p} \phi_i L^i](1 - L)^d X_t = [1 + \sum_{i=1}^{q} \theta_i L^i]\varepsilon_t \tag{1}$$

where: $p$ refers to the number of auto-regression parameters, while $d$ to the order of differencing required for producing stationarity, $q$ to the number of moving average parameters, $t$ to the time unit, $L$ to the lag operator, $\phi(L)$ to stationary auto-regressive operator, $\theta(L)$ to reversible moving average operator, and $d \in z$ to target variable, respectively [39].

Both Auto-regressive and Moving Average models require stationary data, and the mean as well as variance of the time series being constant over time. The constant mean assumption means no cycles or trends in the data, while the constant variance assumption resembles the kind existed in the homogeneity-of-variance of linear regression. The order of differencing refers to the number of times in which each previous observation is subtracted from successive

one until no systematic decrease or increase at the level of series remains while drifting. The noise in the time series drifts up and down across time.

Historic yields of rice, wheat, maize or soybean worldwide from 1961 to 2019 are considered as a time-series variable and can be estimated using ARIMA model as they generally rise over time due to continuous improvement of the inputs into their production by scientific and technical means. In practice, projecting future yields of rice, wheat, maize or soybean worldwide by 2030 is undertaken stepwise as follows: firstly, to produce logarithmic values of specific crop yields from 1961 to 2019 to eliminate the heteroscedasticity, to test the stationarity of time-series and establish 'stationary series' through differencing if not; secondly, to establish such five basic models as ARMA(1,2) model, ARMA(1,1) model, AR(1) model, MA(2) model and MA(1) model simultaneously applied to fit the crop yields from 1961 to 2019 and evaluate the fitness citing RMSE; finally, to apply optimal basic model used for ARIMA (*p,d,q*) modelling to project their future yields between 2020 and 2030, respectively.

In TR model, the year (in ordinal number) is treated as the independent variable while the yield of specific world crop as dependent one. The model with the highest R squared among such five models as Linear, Exponential, Logarithmic, Polynomial and Power, is applied to project future yields between 2020 and 2030 basing the projection on their history since 1961.

Based on the comparison between different yields of specific crop projected respectively by ARIMA model and TR model, resultant yield is selected following RMSE as estimated result of ARIMA-TR model, in which actual yields in 2020 and 2021 are used for testing the reliability of projection, respectively.

**GS model.** In order to accurately predict unknown information through a small amount of known data, Gray System (GS) theory emerged to meet the requirement for their prediction in near future [40]. In this study, the rolling GS model is used for respectively predicting 2-year future yields of rice, wheat, maize and soybean worldwide, in which the prediction is based on their performance in 5-year history, and the formula of time response is represented as follows:

$$\hat{x}^{(1)}(k+1) = \left(x^{(0)}(1) - \frac{b}{a}\right)e^{-ak} + \frac{b}{a} \tag{2}$$

where: *a* refers to developmental gray number whereas *b* to gray number of endogenous control, and *k* to the serial number of value used for the prediction (*k = 1,2. . ., n*); $X^{(0)}$ is original sequence, and $\hat{x}^{(1)}(k+1)$ is predicted value of accumulated sequence.

**Unary regression model.** Individual impacts of global warming on average and top yields of such staple grain crops as rice, wheat, maize and soybean worldwide since 1961 are respectively analyzed using unary regression model in which global mean temperature stands for the independent variable while the yield of crop for dependent one, and such eleven functions as Linear, Logarithmic, Inverse, Quadratic, Cubic, Compound, Power, S, Growth, Exponential and Logistic are simulated simultaneously, compared with each other and selected pursuant to their both R squared and F-statistic values. Similarly, individual impacts of the earth's land precipitation variation on average and top yields of these four staple grain crops worldwide since 1961 are analyzed to further explore how climate change affects them to yield.

## Results

### Projecting average and top yields of four staple grain crops worldwide by 2030

Average and top yields of rice, wheat, maize and soybean worldwide by 2030 are projected using ARIMA-TR model based on their history since 1961, tested by their actual yields in 2020 and 2021 and further validated by GS model, respectively.

**Table 1. Equations of ARIMA-TR models projecting average yields of four staple grain crops worldwide by 2030.**

| Crop | Model | Equation | RMSE |
|---|---|---|---|
| Rice | ARIMA(1,1,1) | $\ln ayr_t = 0.015851 - 0.497232\ln ayr_{t-1} + 1.497232\ln ayr_{t-2} + \varepsilon_t + 0.469079\varepsilon_{t-1}$ | 328.9316 |
| | Polynomial TR | $y = -0.1531x^2 + 60.072x + 1740$ | 79.8150 |
| Wheat | ARIMA(1,1,2) | $\ln ayw_t = 0.019743 + 0.075827\ln ayw_{t-1} + 0.924173\ln ayw_{t-2} + \varepsilon_t + 0.485031\varepsilon_{t-1} - 0.491181\varepsilon_{t-2}$ | 248.9672 |
| | Polynomial TR | $y = -0.0941x^2 + 46.042x + 1055$ | 84.1059 |
| Maize | ARIMA(0,1,1) | $\ln aym_t = 0.019198 + \varepsilon_t - 0.707803\varepsilon_{t-1}$ | 292.9550 |
| | Polynomial TR | $y = 0.2025x^2 + 54.812x + 1919.3$ | 172.1341 |
| Soybean | ARIMA(0,0,1) | $\ln ays_t = 0.015607 + \varepsilon_t - 0.672767\varepsilon_{t-1}$ | 141.1935 |
| | Polynomial TR | $y = 0.0372x^2 + 24.648x + 1163$ | 87.4512 |

Note: In ARIMA model, *ayr* stands for 'average yield of world rice', while *ayw* for 'average yield of world wheat', *aym* for 'average yield of world maize' and *ays* for 'average yield of world soybean'; in TR model, *x* refers to the number of (ordinal) year while *y* to average yields of rice, wheat, maize and soybean at global level, respectively.

## Projecting average yields of four staple grain crops worldwide by 2030

**ARIMA-TR model projecting average yields of four staple grain crops worldwide by 2030.** The equations of ARIMA-TR model projecting average yields of four staple grain crops worldwide by 2030 are shown in Table 1, respectively.

As shown in Table 1, average yields of world rice between 2020 and 2030 projected using polynomial TR model that has a higher R squared than the linear, exponential, logarithmic and power, should be considered as estimated result of ARIMA-TR model according to their values of RMSE, so should these of world wheat, maize and soybean.

ARIMA-TR model projected average yields of four staple grain crops worldwide between 2020 and 2030 are shown in S2 Table of S1 File, respectively.

As shown in S2 Table of S1 File, average yields of world rice between 2020 and 2030 are projected to increase by 8.4% while these of world wheat by 9.7%, these of world maize by 13.7 and these of world soybean by 10.6%. Compared with their actual yields in 2020 and 2021, above projected average yields of world rice are 4.0% higher and 1.5% higher, whereas these of world wheat, world maize and world soybean are 0.1% higher and 0.6% higher, 3.2% higher and 2.3% higher, and 0.3% lower and 2.3% lower, respectively. The differences between actual average yields and projected ones by the ARIMA-TR model are all below 5.0% for these four staple grain crops in 2020 and 2021.

**2. GS model validating the projection of average yields of four staple grain crops worldwide by 2030.** The values of *a* and *b* in GS model predicting average yields of four staple grain crops in 2020 and 2021 are shown in S3 Table of S1 File, respectively. Based on those values of *a* and *b* parameters in S3 Table of S1 File, GS model predicted average yields of four staple grain crops worldwide in 2020 and 2021 are 4689 kg/ha and 4709 kg/ha of rice, 3491 kg/ha and 3504 kg/ha of wheat, 5835 kg/ha and 5815 kg/ha of maize, and 2816 kg/ha and 2842 kg/ha of soybean, respectively. Compared with their actual yields in recent two years, those predicted average yields of world rice are 1.7% higher in 2020 but 1.2% lower in 2021, while those of world wheat are 0.5% higher in 2020 and 0.3% higher in 2021, those of world maize are 1.4% higher in 2020 but 1.1% lower in 2021, and those of world soybean are 1.2% higher in 2020 but 1.0% lower in 2021, respectively. Compared with ARIMA-TR model-projected average yields of four staple grain crops worldwide in recent two years, those predicted by GS model are 2.2% lower in 2020 and 2.6% lower in 2021 for rice, 0.3% higher in 2020 but 0.3% lower in 2021 for wheat, 1.7% lower in 2020 and 3.3% lower in 2021 for maize, and 1.4% higher in 2020 and 1.3% higher in 2021 for soybean, respectively. That is to say, all differences between actual

yields and predicted values by the GS model, and those between ARIMA-TR model-projected and GS model-predicted ones, are also all below 5.0% for average yields of these four staple grain crops, which well validates the projections of ARIMA-TR model.

## Projecting top yields of four staple grain crops worldwide by 2030

Considering that top yields of specific crop worldwide vary much more than its average, whether or not their future can be projected using time-series model depends on if there exists some tendency of increase or decrease in their historic variation. Analyses show that the highest R squared values of variation trend regression of top yields from four staple grain crops worldwide since 1961 are 0.7373 of Polynomial for rice, 0.9349 of Polynomial for wheat, 0.8979 of Polynomial for maize and 0.6405 of Power for soybean, and indicate that these tops all vary in some statistical tendency of increase and then can be used by time-series model for projecting their futures by 2030.

**ARIMA-TR model projecting top yields of four staple grain crops worldwide by 2030.** Likewise, the equations of ARIMA-TR model projecting top yields of four staple grain crops worldwide by 2030 are shown in Table 2, respectively.

As shown in Table 2, top yields of world rice, world wheat and world maize between 2020 and 2030 should be projected using corresponding polynomial TR models (with higher R squared values than the linear, exponential, logarithmic and power) as estimated result of ARIMA-TR model pursuant to their values of RMSE, whereas that of world soybean using power TR model having higher R squared than the other four functions.

Top yields of four staple grain crops worldwide between 2020 and 2030 projected by ARIMA-TR model are shown in S4 Table of S1 File, respectively.

As shown in S4 Table of S1 File, top yields of world rice between 2020 and 2030 are projected to increase by 1.6%, whereas they are world wheat by 1.9%, world maize by 25.4% and world soybean by 3.2%, which indicates that world maize would have the biggest increase range of top yield among four staple grain crops in ensuing decade. Compared with their actual yields in recent two years, above projected top yields of world rice are 0.7% higher in 2020 but 0.7% lower in 2021, whereas they are 0.8% lower and 1.9% lower for world wheat, 23.8% higher and 24.0% higher for world maize, and 7.1% lower and 0.7% lower for world soybean, in 2020 and 2021 respectively. Projected tops of world maize in 2020 and 2021 are obviously higher than their counterparts, mainly for actual yields are relatively low among those since 2013. The differences between actual and ARIMA-TR model projected values of top yields from four staple grain crops (especially world maize) are generally bigger than these of

**Table 2. Equations of ARIMA-TR models projecting top yields of four staple grain crops worldwide by 2030.**

| Crop | Model | Equation | RMSE |
|------|-------|----------|------|
| Rice | ARIMA(1,1,2) | $\ln tyr_t = 0.005414 + 0.053938\ln tyr_{t-1} + 0.946062\ln tyr_{t-2} + \varepsilon_t + 0.673627\varepsilon_{t-1} - 0.209835\varepsilon_{t-2}$ | 1059.049 |
| | Polynomial TR | $y = -0.6739x^2 + 104.09x + 6285.3$ | 170.5170 |
| Wheat | ARIMA(1,1,1) | $\ln tyw_t = 0.013905 + 0.708010\ln tyw_{t-1} + 0.291990\ln tyw_{t-2} + \varepsilon_t - 0.524676\varepsilon_{t-1}$ | 1047.800 |
| | Polynomial TR | $y = -1.2742x^2 + 184.25x + 3388.1$ | 507.2709 |
| Maize | ARIMA(1,1,2) | $\ln tym_t = 0.024757 + 0.493846\ln tym_{t-1} + 0.506154\ln tym_{t-2} + \varepsilon_t + 0.005157\varepsilon_{t-1} - 0.201672\varepsilon_{t-2}$ | 3720.655 |
| | Polynomial TR | $y = 5.8741x^2 + 167.46x + 5516.6$ | 2884.0172 |
| Soybean | ARIMA(0,0,2) | $\ln tys_t = 0.011602 + \varepsilon_t - 0.775173\varepsilon_{t-1} - 0.003830\varepsilon_{t-2}$ | 669.1066 |
| | Power TR | $y = 1764.4x^{0.2062}$ | 520.1776 |

Note: In ARIMA model, *tyr* stands for 'top yield of world rice', while *tyw* for 'top yield of world wheat', *tym* for 'top yield of world maize' and *tys* for 'top yield of world soybean'; in TR model, *x* refers to the number of (ordinal) year while *y* to top yields of world rice, wheat, maize and soybean at national level, respectively.

their averages in both 2020 and 2021 as the latter varied much more smoothly than the former during the same period.

**2. GS model validating the projection of top yields of four staple grain crops worldwide by 2030.** The values of *a* and *b* in GS model predicting top yields of four staple grain crops in 2020 and 2021 are shown in S5 Table of S1 File, respectively. Based on those values of *a* and *b* parameters in S5 Table of S1 File, top yields of four staple grain crops worldwide in 2020 and 2021 predicted by GS model are 9884 kg/ha and 9959 kg/ha of rice, 9772 kg/ha and 10176 kg/ha of wheat, 29894 kg/ha and 30402 kg/ha of maize, and 4260 kg/ha and 4243 kg/ha of soybean, respectively. Compared with their actual yields in recent two years, these predicted top yields of world rice are 1.5% lower in 2020 and 2.4% lower in 2021 while those of world wheat are 1.6% lower in 2020 but 1.0% higher in 2021, those of world maize are 0.8% higher in 2020 and 0.3% higher in 2021, and those of world soybean are 3.6% lower in 2020 but 2.3% higher in 2021, respectively. The differences between actual yields and predicted values by the GS model are also all below 5.0% for top yields of these four staple grain crops, which further validates the projections of ARIMA-TR model. However, compared with ARIMA-TR model-projected top yields of four staple grain crops worldwide in recent two years, those predicted by GS model are 2.2% lower in 2020 and 1.7% lower in 2021 for rice, 0.9% lower in 2020 but 2.9% higher in 2021 for wheat, 18.6% lower in 2020 and 19.1% lower in 2021 for maize, and 3.8% higher in 2020 and 3.0% higher in 2021 for soybean, respectively. The differences between the projections of ARIMA-TR model and the predictions of GS model are below 5.0% for top yields of these staple grain crops except maize in recent two years. In other word, GS model may be more powerful to predict near future of crop yields than ARIMA-TR model used for their projection, but can not project any long term of them.

## Gap between average and top yields of four staple grain crops worldwide by 2030

To explore the gap between maximum national yield and global mean level of respective crop in ensuing decade, individual ratios of average to top yields of four staple grain crops worldwide between 2020 and 2030 are shown in S6 Table of S1 File, respectively.

As shown in S6 Table of S1 File, world soybean will have the highest ratio of average to top yields between 2020 and 2030 whereas world maize will do the least among four staple grain crops. That is to say, given that top yield is considered as potential limit of the average, world maize will have the biggest potential scope of average yield increase among four staple grain crops in that future.

The variation trend of ratio (%) between average and top yields by crop among four staple grains by 2030 is shown in Fig 2, respectively.

As shown in Fig 2, theoretically the variation of ratio (%) between average and top yields by crop over time shows a S-shaped curve trend in which world maize and world wheat are in the left part of turn-point (50.0%) vertical line (or below the horizontal level) while world rice and world soybean in the right part (or above the horizontal level), for the gap between these two yield-kinds will be increasingly narrowed in far future.

The variation trends of average and top yields of four staple grain crops worldwide in 1961 to 2021 and to 2030 are shown in Fig 3, respectively.

As shown in Fig 3, in ensuing decade: the gap between average and top yields of world rice would be slightly narrowed, whereas that of world wheat would be gradually contracted, that of world maize would be greatly widened, and that of world soybean would be obviously shrunk, respectively. Theoretically, the gap between average and top yields of certain crop worldwide may temporarily be widening (like world maize) but will be surely closing in far future as the top eventually rises harder and reaches its limit earlier.

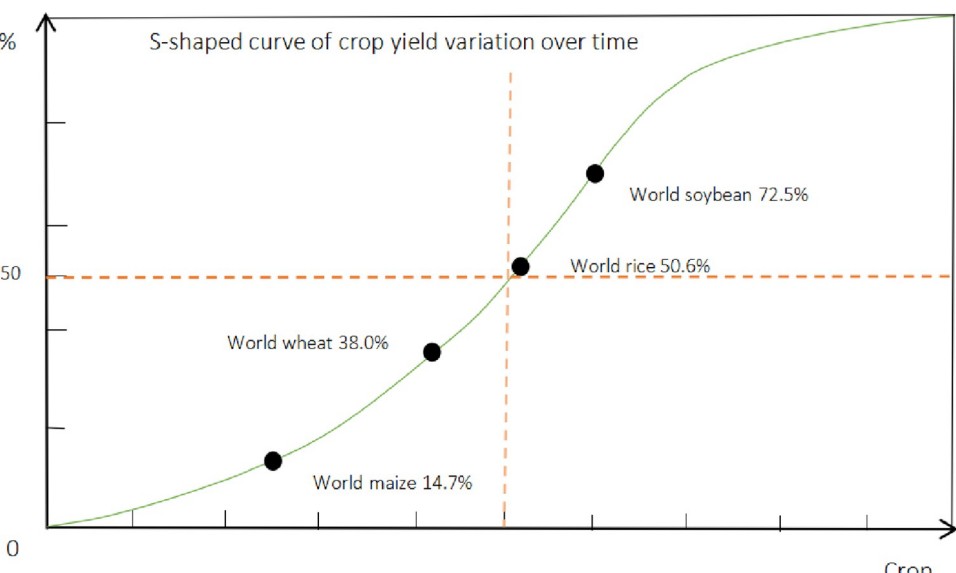

**Fig 2. Variation trend of ratio (%) between average and top yields of four staple grain crops by 2030.**

## Distribution of countries with top yields of four staple grain crops worldwide from 1961 to 2021

Top yields of four staple grain crops from 1961 to 2021 are distributed in these countries with their distribution intensities (viz. the number of years accounting some percentage of totality)

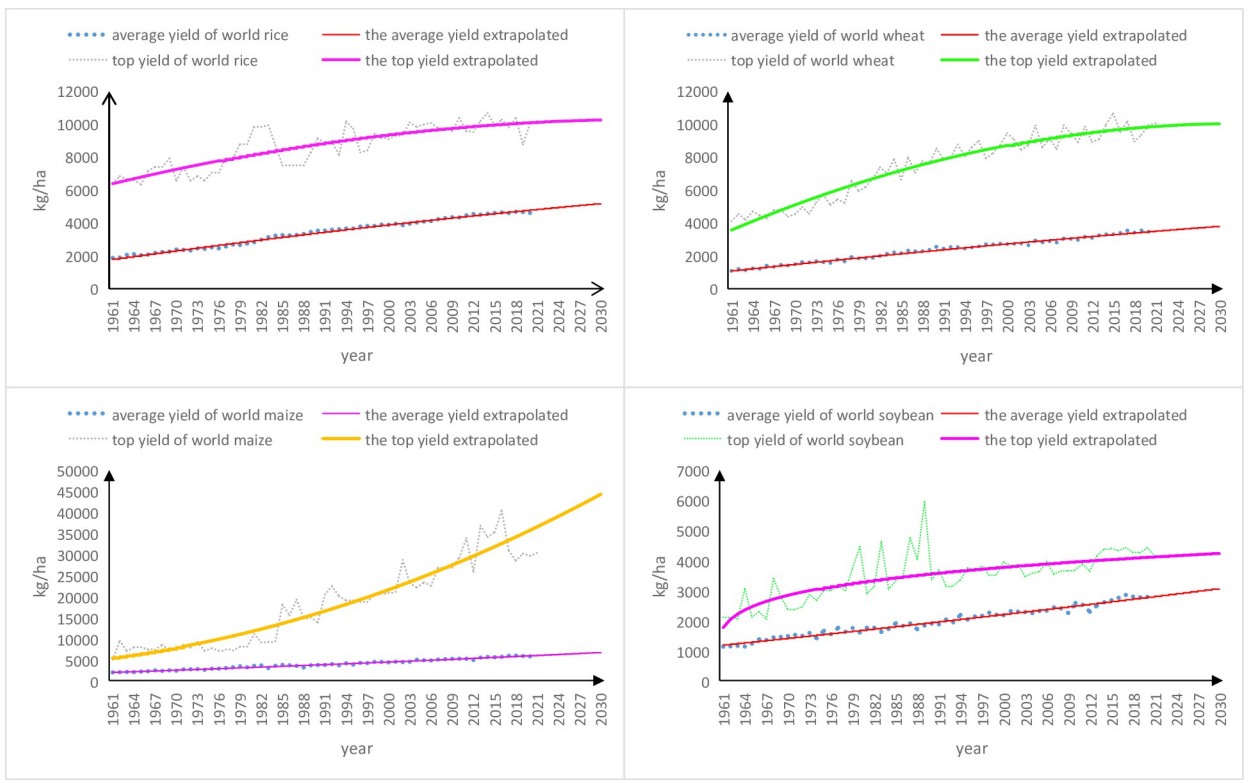

**Fig 3. Variation trends of average and top yields of four staple grain crops worldwide in 1961 to 2021 and to 2030.**

as follows. Top yields of world rice are from these: Dominica, North Korea and Uzbekistan with one year each (1.6%); Syria with two years (3.3%); Swaziland with six years (9.8%); Egypt with thirteen years (21.3%); Puerto Rico with fifteen years (24.6%); and Australia with twenty-two years (36.1%). Top yields of world wheat are in such details: France, Luxembourg and Zambia each with one year (1.6%); Belgium with two years (3.3%); Denmark and New Zealand each with five years (8.2%); Ireland with nineteen years (31.1%); and Netherlands with twenty-seven years (44.3%). Top yields of world maize are so detailed: Qatar with one year (1.6%); Netherlands, Greece, and Saint Vincent plus the Grenadines each with two yearsh (3.3%); Kuwait with three years (4.9%); Israel and Jordan each with six years (9.8%); France with eight years (13.1%); New Zealand with eleven years (18.0%); and United Arab Emirates with twenty years (32.8%). And top yields of world soybean are thus detailed: Canada, Switzerland and Egypt each with one year (1.6%); Mexico and New Zealand each with three years (4.9%); Paraguay with four years (6.6%); Ethiopia PDR with twelve years (19.7%); Turkey with seventeen years (27.9%); and Italy with nineteen years (31.1%). Top yields of four staple grains since 1961 all show intuitively stochastic distribution worldwide, which implies that in a sense various countries having the tops of certain crop in the world constitutes a series of casual events with different probabilities of occurrence.

### Individual impacts of climate change on average and top yields of four staple grain crops worldwide since 1961

Climate change as a 'public bad' worldwide that mainly presents global warming and variation of rainfall, must have some influence on both top and average yields of any world crop.

### Individual impacts of global warming and 'Carbon Peak' target on average and top yields of four staple grain crops worldwide since 1961

Why did individual gaps between average and top yields of four staple grain crops since 1961 vary differently? Theoretically, the variation trend of world crop yield over time in a long run is mainly driven by climate change especially in the form of global warming.

As is well known, annual global mean temperature has been rising in slight fluctuation over time since industrial revolution. And both average and top yields of rice, wheat, maize and soybean worldwide have been rising since 1961 in general. Undoubtedly, there must exist certain inner correlations between global warming and the yields of rice, wheat, maize or soybean worldwide because the temperature is an essential factor for any crop to grow and yield. Though all climatic factors such as sunlight, temperature, precipitation and gases each has respective contribution to the growth and yield of certain crop on global scale, but only the variation (viz. rise) of annual global mean temperature is observed and proved to be the outcome of higher $CO_2$ concentration in atmosphere. And the yield of crop is dependent mainly on climatic factors at global or macroscopic level while fundamentally on nutritional condition and management on local or microscopic scale. At global or macroscopic level of climate change, sunlight and gases though have somehow changed but have primarily made contribution to the yield of crop inclusively in the form of greenhouse effect, for they cause the decrease or increase of global mean temperature directly or indirectly. However, the variation of annual precipitation over time on global scale does not show any obvious trend of increase or decrease. Therefore, firstly for the sake of simplification, the contributions of sunlight, precipitation and gases yearly on global scale to the yield of certain crop worldwide grown in different seasons, are treated as the constant in modelling.

Thus, taking global mean temperature as the independent variable (X) while average and top yields of four staple grain crops worldwide as dependent one (Y), individual impacts of

**Table 3. Model results for individual impacts of global warming on average and top yields of four staple grain crops from 1961 to 2021.**

| Crop | Yield | Equation | Model Summary and Parameter Estimates | | | | | |
|------|-------|----------|----------|----------|----------|----------|----------|----------|
| | | | R Square | F | Constant | b1 | b2 | b3 |
| Rice | Average | Cubic | 0.826 | 137.764 | -51481.040 | 4961.856 | 0.000 | -5.482 |
| | Top | Cubic | 0.683 | 62.445 | -150479.919 | 15628.784 | 0.000 | -21.932 |
| Wheat | Average | Logarithmic | 0.818 | 234.196 | -45337.722 | 17928.460 | | |
| | Top | Cubic | 0.741 | 82.978 | -192200.193 | 19273.526 | 0.000 | -25.879 |
| Maize | Average | Quadratic | 0.827 | 138.207 | 1740.184 | -1785.238.603 | 134.945 | |
| | Top | Quadratic | 0.781 | 103.707 | 115128.073 | -29747.733 | 1601.813 | |
| Soybean | Average | Quadratic | 0.841 | 153.226 | 4255.031 | -1171.840 | 70.650 | |
| | Top | S | 0.442 | 46.700 | 12.436 | -61.573 | | |

Note: all equations are at great significance level of 0.1%.

global warming on their production from 1961 to 2021 are analyzed using unary regression models and shown in Table 3, respectively.

As shown in Table 3, from 1961 to 2021 four staple grain crops are affected by global warming as follow. Both average and top yields of world rice are affected by global warming in cubic function that has higher R-squared/F-statistic values than the linear, logarithmic, inverse, quadratic, compound, power, S, growth, exponential and logistic, whereas average yield of world wheat in logarithmic function but the top in cubic one, both average and top yields of world maize in quadratic function, and average yield of world soybean in quadratic while the top in S function. They are specifically affected in following sense. Average yield of world rice is impacted by global warming negatively less than the top, which partly drives the gap between these two yields slightly narrowed. Average yield of world wheat is affected by global warming positively while the top negatively, which makes the gap between these two yields gradually contracted since the middle period. In a way, the reason for different effects of global warming on average and top yields among these two most important cereal crops is that rice is predominantly grown in tropical and subtropical zones whereas wheat pervasively in subtropical and temperate ones. Average yield of world maize rises with global warming less than the top, which drives the gap between these two yields greatly widened. This is further proved that maize is evolved later in biological evolution and more heat-endurable to some extent than rice and wheat among three staple cereals, and has the biggest potential scope of average yield increase among four staple grains under global warming. And average yield of world soybean progresses is affected by global warming positively while the top negatively, which makes the gap between these two yields obviously shrunk after the first short period.

Since the 'Carbon Peak' target agreed in *The Paris Agreement*, most countries in the world have been undertaking various measures to reduce domestic carbon emission. For instance, China promises to try to achieve dual carbon targets of 'Carbon Peak' by 2030 and 'Carbon Neutrality' by 2060, respectively. In this case, the impacts of global warming on average and top yields of four staple grain crops worldwide would be mitigated to some extent.

Global mean temperature from 1961 to 2021 varies in Polynomial trend which has a higher R-squared (0.8748) than Linear (0.8481), Exponential (0.8479), Logarithmic (0.5589) and Power (0.5621), and is predicted to be 15.7°C by 2030. Noticeably, global mean temperature by 2030 can not be projected by using ARIMA-TR model of time-series approach because its rise over time is not a stochastic process but largely caused by human activity. In comparison with global mean temperature prior to industrial revolution (13.8°C), global warming is targeted between 1.5°C and 2.0°C in *The Paris Agreement*. Therefore, global warming will

probably reach that ceiling (13.8+2.0 = 15.8°C) soon after 2030 if current tendency kept on without effective human intervene or adjustment of behavior. Namely, global warming could be probably retarded by 2.5% [(15.7–13.8–1.5)/15.7] to the maximum only under 'Carbon Peak' target. As a consequence, compared with the projections of ARIMA-TR model, both average and top yields of world rice might be enhanced, average yield of world wheat might be decreased but its top might be increased, both average and top yields of world maize might be lessened, and average yield of world soybean might be reduced whereas its top might be risen, respectively to the extent that global warming retarded.

Individual impacts of global warming on four staple grain crops to yield from 1961 to 2021 and to 2030 are further analyzed through using unary regression models and shown in S7 Table of S1 File, respectively.

As shown in Fig 3 and S7 Table in S1 File, individual impacts of global warming on average and top yields of rice, wheat, maize and soybean worldwide between two periods of 1961–2021 and 1961-2021-2030 are generally consistent in terms of tendency.

## Individual impacts of global rainfall on average and top yields of four staple grain crops worldwide since 1961

To further explore individual impacts of climate change on the production of four staple grain crops, similarly taking land precipitation of the earth as the independent variable (X) while average and top yields of four staple grain crops worldwide as dependent one (Y), the effects of global rainfall on them to yield from 1961 to 2021 are similarly analyzed using unary regression models and shown in Table 4, respectively.

As shown in Table 4, from 1961 to 2021 both average and top yields of four staple grain crops are all positively affected by global rainfall with much lower R squared and F-statistic values than those of global warming, which indicates that the impacts of land precipitation variation on the production of these grains are not so significant as those of global mean temperature during same period.

Theoretically, the variation of global rainfall is caused partly by global warming. Annual precipitation on land surface of the earth from 1961 to 2021 varies in Polynomial trend with a higher R-squared of 0.0988 than Linear of 0.0763, Exponential of 0.0722, Logarithmic of 0.0314 and Power of 0.0291 that are much lower than these of global warming, which proves that unlike global mean temperature, land precipitation just varied in fluctuation without obvious increasing tendency during the same period. Basing the polynomial trend, annual land precipitation is predicted to be 910 mm by 2030, and the impacts of global rainfall on average and top yields of four staple grain crops between two periods of 1961–2021 and 1961-2021-

**Table 4. Model results for individual impacts of global rainfall on average and top yields of four staple grain crops from 1961 to 2021.**

| Crop | Yield | Equation | Sig. | Model Summary and Parameter Estimates | | | | | |
|---|---|---|---|---|---|---|---|---|---|
| | | | | R Square | F | Constant | b1 | b2 | b3 |
| Rice | Average | Quadratic | 0.001 | 0.210 | 7.708 | 266789.548 | -608.737 | 0.351 | |
| | Top | Cubic | 0.041 | 0.104 | 3.368 | 89523.412 | 0.000 | -0.326 | 0.000 |
| Wheat | Average | Quadratic | 0.001 | 0.210 | 7.728 | 220913.484 | -504.431 | 0.291 | |
| | Top | Quadratic | 0.002 | 0.189 | 6.773 | 560701.449 | -1278.358 | 0.738 | |
| Maize | Average | Cubic | 0.001 | 0.211 | 7.738 | 117557.277 | 0.000 | -0.456 | 0.000 |
| | Top | Quadratic | 0.001 | 0.207 | 7.559 | 2994246.425 | -6862.041 | 3.591 | |
| Soybean | Average | Cubic | 0.000 | 0.236 | 8.968 | 51436.914 | 0.000 | -0.198 | 0.000 |
| | Top | Quadratic | 0.025 | 0.119 | 3.929 | 204901.913 | -462.050 | 0.265 | |

2030 (shown in S8 Table of S1 File) are generally in consistent tendency. In general, global warming contributes to the production of four staple grain crops far more than the variation of land rainfall, and become main force that drives the gap between their average and top yields varied differently over time.

## Discussion

### Two kinds of model respectively based on the principle of production function and the theory of stochastic process

Crop models have been proven as effective tools for supporting research and development endeavors aimed at maximizing agricultural productivity particularly in developing countries, and facilitated the prediction of crop phenology and yield based on various factors influencing growth and development of crop [41]. Typically, crop models are developed and validated by combining data sets and simulations that encompass factors such as environmental condition, soil characteristics, management practice, and crop genotype [42]. A huge number of scholars have made considerable efforts in accounting for as many influential factors as possible when constructing crop models of yield estimation. However, it is worth noting that simpler models can sometimes outperform more advanced counterparts, especially in the situations where limited data are available to calibrate the parameters of model, as intricate interrelationships among influential factors within a system often present insurmountable challenges in accurately identifying and modelling their complexities. For example, Yu et al. (2023) conducted an assessment of three individual crop models, namely AquaCrop, WOFOST and Oryza version 3 (OryzaV3), as well as five multiple model averaging (MMA) methods for predicting regional rice yields during early and late seasons, and found that the crop models with simpler structures exhibited superior performance [43]. In a similar vein, Pham et al. (2023) employed simple crop models to investigate the responses of soybeans to climate conditions in the Mekong Delta of Vietnam, which demonstrated good performance in simulating the yields with overall relative RMSE of 9.0–10.0% [44]. Consistently, our study reveals that, in most cases, relatively simple TR model outperforms ARIMA model to project average and top yields of four staple grain crops, as evidenced by smaller RMSE values.

Rather than most studies that estimate crop yield using models based on the production function in which various contributors (e.g. light, temperature, water, nutrition and gases) are respectively taken into account, this research estimates the yields of four staple grain crops worldwide by 2030 using time-series models in which all influential factors are integrated into one independent. This approach based on the theory of stationary stochastic process depends on enough number of samples or size of scope that conceals some inevitable law behind lots of casual events. The time-series model has the advantage to simplify and integrate complex linkages existed in metabolic system of plant, being particularly applicable to macroscopic and dynamic scenario in which the variable varies on large scale and at long period, just like the variation of world crop yield over time. The wider as well as longer the coverage is, the more accurate the estimation would be. For example: Atamanyuk et al. (2023) pointed out that considering stochastic features of wheat production (e.g. technological and weather parameters) allows the researchers to achieve the best accuracy when predicting the crop yields [45]; Chang et al. (2023) used data-driven and process-based model incorporating machine learning to predict maize yield from available historical data over both temporal and spatial dimensions in the US Corn Belt region from 1981 to 2020, and achieved an accurate prediction performance with a 7.2% relative RMSE of average yield [46]. On the contrary, the models based on the theory of production function are generally better applicable to microscopic and static scenario,

in which the smaller the coverage is the more accurate the estimation will be. These two categories of methodology should not contradict but complement each other.

## Potential yields of different crops with their respective biological characteristics

There exist some similarities as well as discrepancies among different crops in terms of both inside and outside characteristics, on which their yields are dependant to some extent. For instance, rice and wheat are similar in the aspect of plant shape, grain component and C3 metabolism, and then they two share resembling potential limit of average yields worldwide like 10269 kg/ha and 10042 kg/ha by 2030, respectively. On the contrary, maize has C4 metabolism, much taller plant than rice and wheat and then has far higher potential limit of average yield worldwide like 46022 kg/ha by 2030. Compared with these three cereals whose grains largely contain starch, soybean produces more nutritional seed mainly consisted of protein and oil that consumes more solar energy in the process of photosynthesis, and then has a much smaller potential limit of average yield worldwide like 4237 kg/ha by 2030.

Since ancient time, human being has been working hard for seeking high yield of four staple grain crops on the purpose of food security. Through advanced technologies in variety cultivation and genetic breeding, the crop yield can be improved. And genetic engineering is deemed as the most effective way for improving potential yield of crop. Improving seeds through breeding and advanced cultivation should be simultaneously applied to maximize crop yield. However, no matter how advanced seed improving approaches (for example cloning gene) or cultivation technologies (like computer-controlled temperature and water) are applied, there is evidence that any given crop's yield is limited due to the limitation of solar radiation onto unit acreage (e.g. one hectare) during the period of crop growth. As a consequence, 'low yield' and 'high quality' is often 'bundle-sold', and an indissoluble link inevitably occurs that attempting to maximize crop production means accepting some loss in its quality. Of course, under special conditions, crop top yields may be very high, but will not be sustained for a long period or extended on a large scale. Thus in a long run like the increase of population in an ecosystem, the yield of crop that goes up with time theoretically shows some trend of a S-shaped curve where it is positively accelerated in first half whereas negatively in second half until the acceleration stopped eventually, in which global mean temperature is the most important driving factor among climatic factors globally. Therefore, the priority of input into the production of food worldwide should be differentiated and given to the countries (or regions) where the crop has relatively high potential of yield increase as follows. For the crop (like world maize) with average yield still at low stage before the turn-point of such S-shaped curve (e.g. below 30.0% of potential limit), the importance of improving global production should be mainly attached to high-yield countries (or regions) with high efficiency; while for the crop (like world soybean) with average yield already at high stage after the turn-point of such S-shaped curve (e.g. above 70.0% of potential limit), the importance should be mainly attached to low-yield countries (or regions) through the amelioration of arable land and the improvement of management with high input and output; and for the crop (like world rice and world wheat) with average yield in main body of such S-shaped curve (e.g. between 30.0% and 70.0% of potential limit), the importance for improving global production should be attached to both high-yield and low-yield countries (or regions) with integrated efficiency.

## Limitations of this research

Due to the difficulty to obtain corresponding data, this research did not simultaneously consider individual impacts of sunlight, temperature, rainfall, gases, fertilizers, global

desertification and the like on average and top yields of four staple grain crops worldwide, which limits the results to be completely close to the reality in a sense.

## Conclusions

Top yield of world crop could be considered (regionally attainable) as potential limit of its average because in theory the latter will 'chase after' but never meet the former. And the further average yield of world crop lags behind its potential limit, the easier it can be theoretically enhanced; vise versa. Therefore, global food security should be mainly dependant on optimizing sustainable production of crops with larger potential space of average yield increase.

As is projected in this research, the gap between average and top yields of rice will be slightly narrowing, while that of wheat will be gradually closing, that of maize will be greatly widening, and that of soybean will be obviously shrinking, worldwide in ensuing decade. Since 1961, global warming has been exerting a negative impact on average yield of world rice less than on its top, a positive effect on average yield of world wheat while a negative one on its top, a positive effect on average yield of world maize less than on its top, and a positive influence on average yield of world soybean while a negative one on its top. The impacts of global warming on potential yields of four staple grain crops could be slightly mitigated in a way only by effective human management under 'Carbon Peak' target. The earth's rainfall almost makes a constantly positive contribution to the growth and yield of crop worldwide. These insights offer innovative comprehension to academic community and provide policymakers with supportive information on sustainable production of four staple grain crops for global food security facing the challenge of climate change in the future.

## Supporting information

**S1 File. The supporting information file includes S1 to S8 Tables.**
(DOCX)

## Author Contributions

**Conceptualization:** Chengzhi Cai.

**Data curation:** Linyu LV, Sha WEI.

**Formal analysis:** Chengzhi Cai.

**Investigation:** Chengzhi Cai.

**Methodology:** Lin ZHANG.

**Supervision:** Chengzhi Cai.

**Validation:** Chengzhi Cai.

**Visualization:** Lin ZHANG, Wenfang CAO.

**Writing – original draft:** Chengzhi Cai.

**Writing – review & editing:** Linyu LV, Sha WEI, Wenfang CAO.

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
