## [Decision Letter · Decision Letter 0]

13 Feb 2024

PONE-D-23-35004How Does Climate Change Affect Potential Yields of Four Staple Grain Crops Worldwide by 2030?PLOS ONE

Dear Dr. Cai,

Thank you for submitting your manuscript to PLOS ONE. After careful consideration, we feel that it has merit but does not fully meet PLOS ONE’s publication criteria as it currently stands. Therefore, we invite you to submit a revised version of the manuscript that addresses the points raised during the review process.

We look forward to receiving your revised manuscript.

Kind regards,

ML Dotaniya, Ph.D.

Academic Editor

PLOS ONE

3. PLOS requires an ORCID iD for the corresponding author in Editorial Manager on papers submitted after December 6th, 2016. Please ensure that you have an ORCID iD and that it is validated in Editorial Manager. To do this, go to ‘Update my Information’ (in the upper left-hand corner of the main menu), and click on the Fetch/Validate link next to the ORCID field. This will take you to the ORCID site and allow you to create a new iD or authenticate a pre-existing iD in Editorial Manager. Please see the following video for instructions on linking an ORCID iD to your Editorial Manager account: " ext-link-type="uri" xlink:type="simple">https://www.youtube.com/watch?v=_xcclfuvtxQ".

4. Please amend the manuscript submission data (via Edit Submission) to include author LV Linyu, WEI Sha, ZHANG Lin and CAO Wenfang.

5. We note that Figure 4 in your submission contain [map/satellite] images which may be copyrighted. All PLOS content is published under the Creative Commons Attribution License (CC BY 4.0), which means that the manuscript, images, and Supporting Information files will be freely available online, and any third party is permitted to access, download, copy, distribute, and use these materials in any way, even commercially, with proper attribution. For these reasons, we cannot publish previously copyrighted maps or satellite images created using proprietary data, such as Google software (Google Maps, Street View, and Earth). For more information, see our copyright guidelines: http://journals.plos.org/plosone/s/licenses-and-copyright.

a. You may seek permission from the original copyright holder of Figure 4 to publish the content specifically under the CC BY 4.0 license.  

Reviewers' comments:

Reviewer's Responses to Questions

**Comments to the Author**

1. Is the manuscript technically sound, and do the data support the conclusions?

Reviewer #1: Yes

2. Has the statistical analysis been performed appropriately and rigorously? 

Reviewer #1: Yes

3. Have the authors made all data underlying the findings in their manuscript fully available?

Reviewer #1: Yes

4. Is the manuscript presented in an intelligible fashion and written in standard English?

Reviewer #1: Yes

5. Review Comments to the Author

Reviewer #1: Well written article. Corrections indicated in the attached manuscript need to be carried out and submitted again for review. Further, how the results obtained through various models using only few parameters namely temperature, precipitation and yield (average and potential) are going to be close to the reality that too with dwindling natural resources mainly the water and fertilizers and also deteriorating soil health across the globe.

6. PLOS authors have the option to publish the peer review history of their article (what does this mean?). If published, this will include your full peer review and any attached files.

Reviewer #1: No

---

## [Author Response · Author response to Decision Letter 0]

22 Mar 2024

Response to Reviewers

Corresponding author’s response: My manuscript has been formatted to meet PLOS ONE's style requirements, including those for file naming.

Corresponding author’s response: Not applicable.

3. PLOS requires an ORCID iD for the corresponding author in Editorial Manager on papers submitted after December 6th, 2016. Please ensure that you have an ORCID iD and that it is validated in Editorial Manager. To do this, go to ‘Update my Information’ (in the upper left-hand corner of the main menu), and click on the Fetch/Validate link next to the ORCID field. This will take you to the ORCID site and allow you to create a new iD or authenticate a pre-existing iD in Editorial Manager. Please see the following video for instructions on linking an ORCID iD to your Editorial Manager account: https://www.youtube.com/watch?v=_xcclfuvtxQ".

Corresponding author’s response: My ORCID iD is 0000-0002-4772-2495.

4. Please amend the manuscript submission data (via Edit Submission) to include author LV Linyu, WEI Sha, ZHANG Lin and CAO Wenfang.

Corresponding author’s response: These manuscript submission data are amended while submitting the revision.

5. We note that Figure 4 in your submission contain [map/satellite] images which may be copyrighted. All PLOS content is published under the Creative Commons Attribution License (CC BY 4.0), which means that the manuscript, images, and Supporting Information files will be freely available online, and any third party is permitted to access, download, copy, distribute, and use these materials in any way, even commercially, with proper attribution. For these reasons, we cannot publish previously copyrighted maps or satellite images created using proprietary data, such as Google software (Google Maps, Street View, and Earth). For more information, see our copyright guidelines: http://journals.plos.org/plosone/s/licenses-and-copyright.

a. You may seek permission from the original copyright holder of Figure 4 to publish the content specifically under the CC BY 4.0 license. 

Natural Earth (public domain): http://www.naturalearthdata.com/.

Corresponding author’s response: Figure 4 has been removed.

Corresponding author’s response: The captions for my Supporting Information files have been included at the end of my manuscript, and the in-text citations have been updated to match accordingly.

Corresponding author’s response: My reference list has been reviewed and corrected.

Reviewers' comments:

Reviewer's Responses to Questions

Comments to the Author

1. Is the manuscript technically sound, and do the data support the conclusions?

Reviewer #1: Yes

2. Has the statistical analysis been performed appropriately and rigorously?

Reviewer #1: Yes

3. Have the authors made all data underlying the findings in their manuscript fully available?

Reviewer #1: Yes

4. Is the manuscript presented in an intelligible fashion and written in standard English?

Reviewer #1: Yes

5. Review Comments to the Author

Reviewer #1: Well written article. Corrections indicated in the attached manuscript need to be carried out and submitted again for review. Further, how the results obtained through various models using only few parameters namely temperature, precipitation and yield (average and potential) are going to be close to the reality that too with dwindling natural resources mainly the water and fertilizers and also deteriorating soil health across the globe.

Corresponding author’s response: The corrections have been carried out as indicated. In addition, due to the difficulty to obtain corresponding data, this research did not simultaneously consider individual impacts of sunlight, temperature, rainfall, gases, fertilizers, global desertification and the like on average and top yields of four staple grain crops worldwide, which limits the results to be completely close to the reality in a sense. This limitation of research is amended in Discussion section accordingly.

6. PLOS authors have the option to publish the peer review history of their article (what does this mean?). If published, this will include your full peer review and any attached files.

Do you want your identity to be public for this peer review? For information about this choice, including consent withdrawal, please see our Privacy Policy.

Reviewer #1: No.

---

## [Editor Report · Decision Letter 1]

2 May 2024

How Does Climate Change Affect Potential Yields of Four Staple Grain Crops Worldwide by 2030?

PONE-D-23-35004R1

Dear Dr. Cai,

We’re pleased to inform you that your manuscript has been judged scientifically suitable for publication and will be formally accepted for publication once it meets all outstanding technical requirements.

Kind regards,

ML Dotaniya, Ph.D.

Academic Editor

PLOS ONE
---

## [Editor Report · Acceptance letter]

8 May 2024

PONE-D-23-35004R1 

PLOS ONE

Dear Dr. Cai, 

I'm pleased to inform you that your manuscript has been deemed suitable for publication in PLOS ONE. Congratulations! Your manuscript is now being handed over to our production team.

Kind regards, 

on behalf of

Dr. ML Dotaniya 

Academic Editor

PLOS ONE